# Effect of HIV status and antiretroviral treatment on treatment outcomes of tuberculosis patients in a rural primary healthcare clinic in South Africa

**Peter S. Nyasulu**[1,2]\*, **Emery Ngasama**[2], **Jacques L. Tamuzi**[1], **Lovemore N. Sigwadhi**[1], **Lovelyn U. Ozougwu**[3], **Ruvimbo B. C. Nhandara**[2], **Birhanu T. Ayele**[1], **Teye Umanah**[4], **Jabulani Ncayiyana**[5]

**1** Division of Epidemiology and Biostatistics, Department of Global Health, Faculty of Medicine and Health Sciences, Stellenbosch University, Cape Town, South Africa, **2** Division of Epidemiology & Biostatistics, School of Public Health, Faculty of Health Sciences, University of the Witwatersrand, Johannesburg, South Africa, **3** Division of Public Health Surveillance and Response, National Institute for Communicable Diseases, Johannesburg, South Africa, **4** Neuroscience Institute, Hackensack Meridian University at JFK Medical Center, Hackensack, New Jersey, United States of America, **5** Public Health Medicine, School of Nursing and Public Health, University of KwaZulu-Natal, Durban, South Africa

\* pnyasulu@sun.ac.za

**Data Availability Statement:** All relevant data are within the paper and its Supporting information files.

## Abstract

### Background

Tuberculosis (TB) remains the leading cause of death among human immunodeficiency virus (HIV) infected individuals in South Africa. Despite the implementation of HIV/TB integration services at primary healthcare facility level, the effect of HIV on TB treatment outcomes has not been well investigated. To provide evidence base for TB treatment outcome improvement to meet End TB Strategy goal, we assessed the effect of HIV status on treatment outcomes of TB patients at a rural clinic in the Ugu Health District, South Africa.

### Methods

We reviewed medical records involving a cohort of 508 TB patients registered for treatment between 1 January 2013 and 31 December 2015 at rural public sector clinic in KwaZulu-Natal province, South Africa. Data were extracted from National TB Programme clinic cards and the TB case registers routinely maintained at study sites. The effect of HIV status on TB treatment outcomes was determined by using multinomial logistic regression. Estimates used were relative risk ratio (RRR) at 95% confidence intervals (95%CI).

### Results

A total of 506 patients were included in the analysis. Majority of the patients (88%) were new TB cases, 70% had pulmonary TB and 59% were co-infected with HIV. Most of HIV positive patients were on antiretroviral therapy (ART) (90% (n = 268)). About 82% had successful treatment outcome (cured 39.1% (n = 198) and completed treatment (42.9% (n = 217)), 7% (n = 39) died 0.6% (n = 3) failed treatment, 3.9% (n = 20) defaulted treatment and the rest

**Funding:** The author(s) received no specific funding for this work.

**Competing interests:** The authors have declared that no competing interests exist.

(6.6% (n = 33)) were transferred out of the facility. Furthermore, HIV positive patients had a higher mortality rate (9.67%) than HIV negative patients (2.91%)". Using completed treatment as reference, HIV positive patients not on ART relative to negative patients were more likely to have unsuccessful outcomes [RRR, 5.41; 95%CI, 2.11–13.86].

## Conclusions

When compared between HIV status, HIV positive TB patients were more likely to have unsuccessful treatment outcome in rural primary care. Antiretroviral treatment seems to have had no effect on the likelihood of TB treatment success in rural primary care. The TB mortality rate in HIV positive patients, on the other hand, was higher than in HIV negative patients emphasizing the need for enhanced integrated management of HIV/TB in rural South Africa through active screening of TB among HIV positive individuals and early access to ART among HIV positive TB cases.

## 1. Background

Tuberculosis (TB) is one of the major global infectious diseases with a high morbidity and mortality rate. Globally, an estimated 10.0 million (range 8.9–11.0 million) people fell ill with TB in 2019 [1]. There were 1.2 million (range, 1.1–1.3 million) TB deaths among human immunodeficiency virus (HIV)—negative people and an additional 208 000 (range, 177 000–242 000) TB deaths among HIV-positive people [1]. Ninety percent (90%) of all TB cases occur in adults, with a high prevalence in males compared to females [2]. The increase in incidence is also attributed to the development of multidrug resistant (MDR) and extremely drug resistant (XDR) strains of *Mycobacterium tuberculosis* [3]. Both MDR and XDR-TB are the causes of high TB mortality [4]. HIV prevalence in sub-Saharan Africa has significantly contributed to an increase in the incidence of TB [5]. An estimated 10 million people have been living with TB and 1.5 million have died in 2018 and the global burden of TB falls on 20 low-and middle-income (LMIC) countries, including sub-Saharan Africa [6]. The burden is attributed to the high rate of HIV/ AIDS pandemics in those countries. Africa accounted for 84% of all TB/HIV deaths [6].

TB is the most common opportunistic infection and cause of death among PLWHIV in resource-limited countries [7, 8]. In the post-mortem period, the overall prevalence of TB was enormous and accounted for almost 40% of HIV-related facility-based deaths in resource-limited countries such as South Africa [9–11], Botswana [11, 12], Zimbabwe [11, 13, 14], Mozambique [11, 15], Uganda [11, 16] and Kenya [11, 17]. In contrast, the WHO reported 16% of HIV/TB related death in Africa [1].

South Africa (SA) ranked fifth among the highest TB incidences worldwide and first among TB/HIV co-infection cases with more than 65 percent of patients co-infected with TB/HIV [5]. TB accounted for the third highest number of deaths in 2018 (6%; n = 454,014), and combined TB and HIV accounted for 35.6% of all-cause mortality in SA [18–20]. The total number of people living with HIV in SA increased from an estimated 4.64 million in 2002 to 7.97 million by 2019 [21]. There are significant geographical variations in the rate of TB notification in SA which are not clearly correlated with the prevalence of HIV at the district level [21–23]. KwaZulu-Natal (KZN) carries the largest burden of HIV and related infections in SA, with HIV–TB co-infection estimated at approximately 70% [5, 24], and Ugu district reported 60.5% of TB/HIV co-infection rate [25].

According to recent data, 90% of people are aware of their HIV status of which 68% are on antiretroviral therapy (ART) and of which 87% are virally suppressed in SA [26]. ART may be associated with a reduced risk of HIV-associated TB disease in HIV-positive individuals due to a decrease in their viral load and an improvement in their immune system function [27]. On the one hand, while ART reduces new HIV infections, on the other hand, the marked decline in HIV-associated mortality has led to an increase in HIV prevalence and an increase in the number of life-years at TB risk. It is also plausible that HIV-positive people with TB infection may increase with CD4+ T-cell ART recovery, although the available data do not support this [27–29]. In addition, early initiation of ART during TB treatment (within 2–4 weeks) increased AIDS-free survival by 34–68% among patients with advanced HIV disease [11, 30–32]. Despite the inclusion of TB-HIV in the international and SA guidelines, the 2018 mortality rate for co-infected TB-HIV patients in SA was 73 (51–99) per 100,000 populations, which is more than three times higher than that for HIV-negative TB patients with a mortality rate of 37 (35–39) per 100,000 population [33]. TB incidence and mortality are declining in SA [7]. Data from a well-characterized rural SA population with high HIV prevalence and TB incidence demonstrated considerable spatial heterogeneity in people with recently diagnosed TB, and has shown that every percentage increase in ART coverage was associated with a 2% decrease in the odds of recently diagnosed TB [23].

The End TB Strategy aims to reduce TB deaths by 90% and TB incidence rates by 80% in 2030 compared to 2015 [34]. Examining the various challenges that may impact on TB outcomes in rural primary health care in SA, the TB elimination target set for 2050 could be compromised if this dual burden of TB and HIV diseases is not controlled [5, 35]. The rate of TB incident stabilizes at a rate higher than that of the general population. These data highlight the need for more research into strategies for finding active cases in rural settings and the need to focus on strengthening primary health care [36]. Therefore, this retrospective study has been undertaken to determine TB outcomes in HIV positive patients in rural primary healthcare in Ugu Health District, KwaZulu-Natal, SA.

## 2. Methods

### 2.1. Study design

This retrospective cohort study of TB patients initiated on TB treatment was conducted from 1 January 2013 to 31 December 2015.

### 2.2. Study population

We included all patients diagnosed with TB irrespective of their age and HIV status in the study. Further, we included individuals that were registered as TB patients in 2012 and completed treatment or died in 2013. We also included as well as those that died or survived during the treatment period. Patients with unknown outcome or those with incomplete record were excluded Basic demographic information including the age, gender, co-morbidities, tobacco Use, alcohol Use, substance use and duration on treatment were collected.

### 2.3. Study setting

The Ugu Health District is in the rural aspects of the KwaZulu-Natal (KZN) province (Figs 1 & 2). Ugu district has a population of 733 228 people [37]. During the study period, Ugu district had the highest HIV prevalence and TB incidence of any district in KZN, 41.7% and 1096 per 100,000 people, respectively [37]. In terms of infectious TB (pulmonary smear-positive), Ugu ranks 12th, with 325 cases per 100,000 people, which is higher than the country's average of

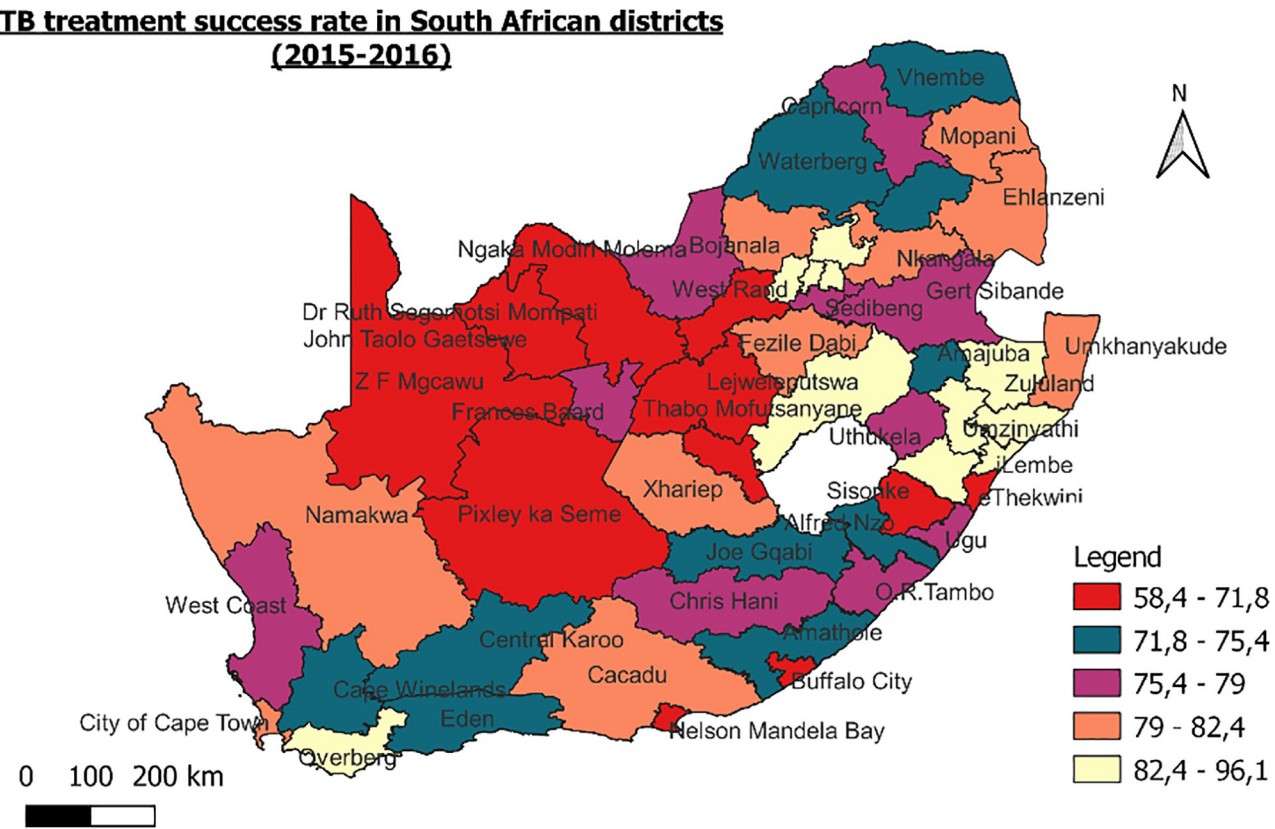

**Fig 1. TB treatment success rate in South African districts.**

208 cases per 100,000 people [37]. Elim clinic, a primary health care facility was selected based on convenience, the study goal, and the availability of information on HIV and TB infections".

## 2.4. Sample size

A total of 478 cases were estimated to be adequate sample that would have the power to detect a significant difference in mortality between HIV positive and negative TB patients on anti-tuberculosis treatment. We used a 95% precision of estimate with power of 80% and estimated risk difference of 6% in outcome between exposed and unexposed.

## 2.5. Participant selection and definitions

All TB patients of all ages recorded in the TB treatment register, with HIV positive status and started on treatment were included in this study. These were all TB patients on treatment attending care at Elim Clinic, in Ugu District, KZN province within the stipulated time. At admission, patients were screened for other comorbidities including HIV infection. After following the treatment, the outcomes of treatment were recorded in seven mutually exclusive categories. Patients were classified as cured, completed treatment, interrupted treatment, moved, transferred out, failed, and died. The main interest of this study was to compare the likelihood of observing two main outcomes. These included being cured versus all other outcomes including death and the likelihood of dying versus staying alive regardless of the outcome of treatment. The World Health Organization (WHO) defines "cured" as follows: a pulmonary TB patient with bacteriologically confirmed TB at the start of treatment who was smear or culture negative

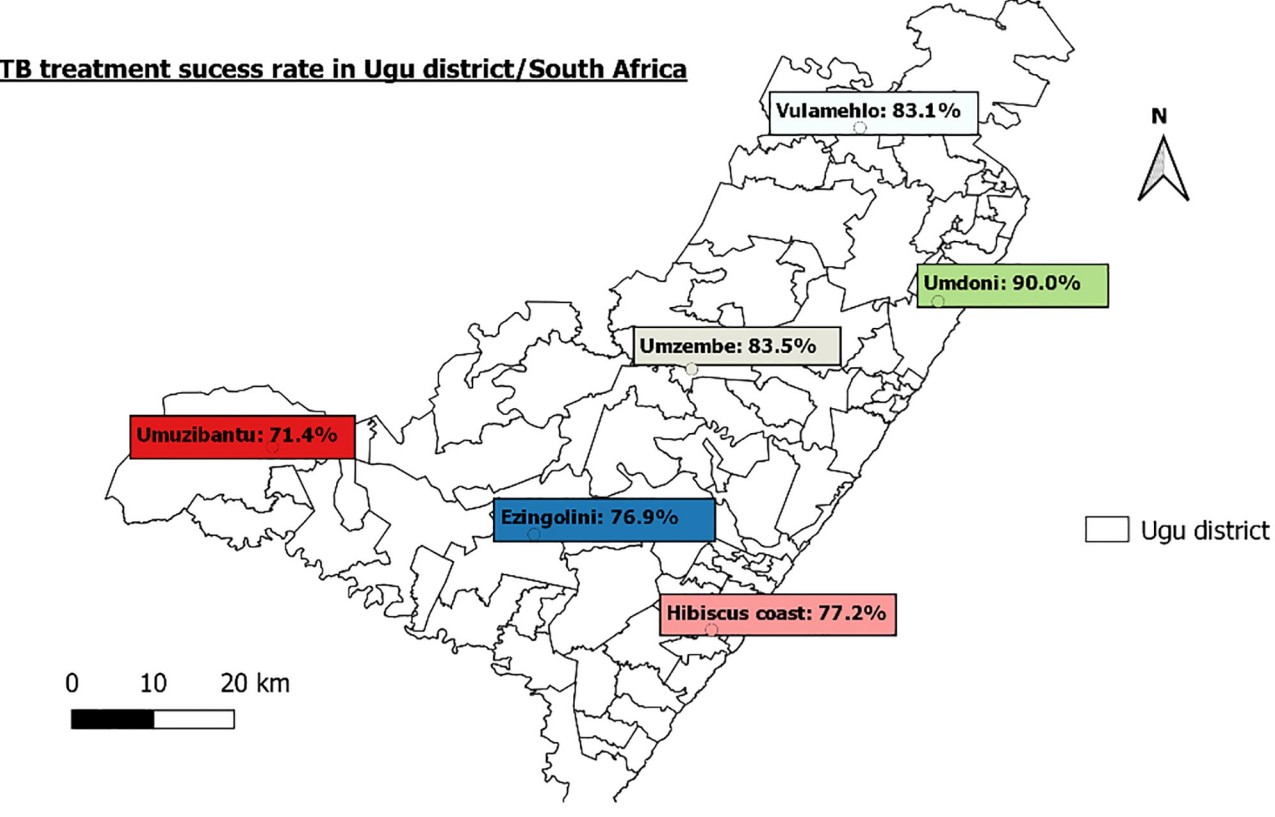

**Fig 2. TB treatment success rate in Ugu district/South Africa.**

in the last month of treatment and on at least one previous occasion, and "treatment completed": A TB patient who completed treatment without evidence of failure but with no record to show that sputum smear or culture results in the last month of treatment and on at least one previous occasion were negative either because tests were not done or because results are unavailable [38]. Treatment success includes the sum of cured and treatment completed [38].

## 2.6. Outcome and exposures

The first outcome of interest was "treatment success" defined as 1 if the patient was declared "treatment success" by the healthcare team or 0 if otherwise. The second outcome was "death" coded 1 if the patient was declared dead and 0 if otherwise. Given the small number of deaths which were 35 in total, we created a nominal outcome, which captured all outcomes and combined them into 3 categories: treatment success, dead and other outcomes.

## 2.7. Data collection

Data were extracted from the TB register using structured data collection form from all the TB patients initiated on treatment at this rural public health clinic from 1 January 2013 and 31 December 2015. The data consists of collecting socio-demographic factors (age and gender), TB related factors (categories, pre-treatment smear, type, site, outcome duration, regimen at the baseline) and other factors such as ART initiation and co-trimoxazole prophylaxis, comorbidities, tobacco, and alcohol use. A data dictionary was formulated to guide data cleaning and analysis. No personal identifiers were extracted from the manual medical records.

## 2.8. Data management

Data were entered on excel spread sheet and imported into Stata software. This step was followed by thorough checking to assess missing values. Data were coded into numerical values and continuous data categorised; clean data were imported into Stata version 15 for analysis.

## 2.9. Data analysis

The main interest of this study was to compare the likelihood of observing two main outcomes. These included being cured versus all other outcomes including death and the likelihood of dying versus staying alive regardless of the outcome of treatment. Data were summarized and presented in tables. As part of the descriptive statistics for the population under study, summary tables were constructed to describe variables in terms of their frequency distribution. Pearson Chi-square and Fisher Exact tests were used to test association between independent variables and TB outcomes (cured, completed treatment, interrupted treatment, moved, transferred out, failed, and died). Given the fact that cured was a common outcome (39%), we fitted a log-binomial regression model to estimate the relative risk of being declared cured versus not being cured (Table 2). Finally, we fitted a multinomial logistic regression model predicting the likelihood of cured and death versus other outcomes. This allowed us to make some predictions on death, as a function of the selected socio-demographic covariates. Odds ratios were computed to assess the association between HIV status and TB mortality. Precision of estimate of the odds ratio was set at 5% significance level. All factors with $p \leq 0.1$ on univariate analysis were further analysed in multiple logistic regression model.

## 2.10. Ethical considerations

Ethical approval was granted by the Monash University Human Research Ethics Committee and permission to conduct the study and access the medical records at Elim Primary Health Care Clinic was obtained from the District Manager of Ugu Health District, KwaZulu Natal province. Waiver of consent was applied for and obtained since the data were retrieved from older clinic records. Data entry and analysis were done anonymously, and no attempt was made to link data to the individual identifier.

## 3. Results

Overall, the study included 508 patients. Of this total, 2 patients were dropped from further analysis because one was transferred to another facility the same day of admission to the hospital and the other one was on prophylaxis TMP-SMX and was HIV negative. Hence, our final analytical sample included 506 patients for whom we had valid data on the TB treatment outcomes. Among them, only 13% (66/506) were children.

Table 1 reported the demographic characteristics and TB outcomes. The findings revealed that 39.13% (n = 198) were reported as being treatment success at the end of treatment. The remaining 60.87% patients were classified as dead or alive, but not treatment success. Some patients were transferred to other facilities, whereas other were classified as treatment interrupted, transferred to another district, treatment outcome unknown, treatment completed and moved to another facility in the same district. Table 1 shows significant differences with regards to TB treatment success in terms of age, tobacco, and alcohol uses. For instance, younger patients had lower proportion of TB treatment success (22.62% of those less than 20 years of age) compared to older patients. Among tobacco users, we found that about 60% had TB treatment success, whereas among non-tobacco users only 37% had TB treatment success. Likewise, the findings estimated that alcohol users, had higher proportion of being TB

**Table 1. Characteristics of TB patients presenting to Elim Clinic on the basis of TB cured outcomes.**

| Factors | Not Cured n (%) | Cured n (%) | P-value | Alive) n (%) | Dead n (%) | p-value |
|---|---|---|---|---|---|---|
| | 308 (60.87) | 198 (39.13) | | 471 (93.08) | 35 (6.92) | |
| **Duration on treatment**[1] | 5.55 (0.16) | 6.15 (0.13) | 0.009* | 6.01 (0.11) | 2.74 (0.44) | 0.001* |
| **HIV Status** | | | | | | |
| Negative | 123 (59.71) | 83 (40.29) | 0.658 | 200 (97.09) | 6 (2.91) | 0.004* |
| Positive | 185 (61.67) | 115 (38.33) | | 271 (90.33) | 29 (9.67) | |
| **Age-group** | | | | | | |
| <20 | 65 (77.38) | 19 (22.62) | 0.018* | 82 (97.62) | 2 (2.38) | 0.101[F] |
| 20–29 | 78 (58.21) | 56 (41.79) | | 124 (92.54) | 10 (7.46) | |
| 30–39 | 72 (55.38) | 58 (44.62) | | 120 (92.31) | 10 (7.69) | |
| 40–49 | 46 (58.97) | 32 (41.03) | | 75 (96.15) | 3 (3.85) | |
| 50+ | 47 (58.75) | 33 (41.25) | | 70 (87.50) | 10 (12.50) | |
| **Sex** | | | | | | |
| Female | 138 (61.61) | 86 (38.39) | 0.762 | 205 (91.52) | 19 (8.48) | 0.216 |
| Male | 170 (60.28) | 112 (39.72) | | 266 (94.33) | 16 (5.67) | |
| **Comorbidity** | | | | | | |
| No | 286 (60.98) | 183 (39.02) | 0.855 | 438 (93.39) | 31 (6.61) | 0.311[F] |
| Yes | 22 (59.46) | 15 (40.54) | | 33 (89.19) | 4 (10.81) | |
| **Tobacco Use** | | | | | | |
| No | 292 (62.66) | 174 (37.34) | 0.005* | 432 (92.70) | 34 (7.30) | 0.345[F] |
| Yes | 16 (40.00) | 24 (60.00) | | 39 (97.50) | 1 (2.50) | |
| **Alcohol Use** | | | | | | |
| No | 296 (61.92) | 182 (38.08) | 0.045* | 443 (92.68) | 35 (7.32) | 0.246[F] |
| Yes | 12 (42.86) | 16 (57.14) | | 28 (100.00) | 0 (0.00) | |
| **Substance Use** | | | | | | |
| No | 292 (60.08) | 194 (39.92) | 0.101 | 452 (93.00) | 34 (7.00) | 1.00[F] |
| Yes | 16 (80.00) | 4 (20.00) | | 19 (95.00) | 1 (5.00) | |

Note:

[1] Duration of treatment in months

All p-value from Pearson Chi-square test, unless otherwise determined.

* p-value < 0.05;

[F] indicates results from Fisher Exact test

treatment success (57.14% of all alcohol users) while non-alcohol users' patients only (38.08%) were TB treatment success.

The overall TB related mortality rate was 6.92% (n = 35 patients). The mortality rate was significantly associated with HIV status. For instance, the mortality rate was higher among HIV positive patients (9.67%, i.e., 29 patients out of 300 patients) compared to HIV negative patients among whom only for 2.91% was reported (6 patients out of 206 HIV negative patients). Given the small sample size for the mortality rate, this was impractical to perform further analysis and obtain reliable estimates.

Table 2 below presented the results from the univariate and multivariate log-binomial model, which sought to identify covariates associated with cure event. These factors included 'duration of treatment, age, comorbidity, tobacco, alcohol use, and HIV positive status' which was our main exposure of interest.

We found that HIV positive status showed to affect the likelihood of TB cure among patients on TB treatment. The results seem to suggest that HIV positive patients were 12% less

**Table 2. Factors associated with treatment success among TB patients presenting for care at Elim Clinic.**

| Factors | Univariate Analysis RR (95% C.I.) | Adjusted Analysis RR (95% C.I.) | p-value |
|---|---|---|---|
| **Duration on treatment** | **1.02 (1.01–1.05)** | **1.03(1.01–1.06)** | 0.039 |
| **HIV Status** | | | |
| Negative | Ref | Ref | |
| Positive | 0.95 (0.76–1.19) | 0.88(0.69–1.11) | 0.124 |
| **Age-group** | | | |
| <20 | Ref | Ref | |
| 20–29 | 1.85(1.19–2.88) | 1.82(1.15–2.91) | 0.014 |
| 30–39 | 1.97(1.27–3.06) | 2.00(1.26–3.17) | 0.008 |
| 40–49 | 1.81(1.13–2.92) | 1.79(1.10–2.91) | 0.033 |
| 50+ | 1.82(1.13–2.93) | 1.77(1.09–2.85) | 0.036 |
| **Sex** | | | |
| Female | Ref | - | |
| Male | 1.03(0.83–1.29) | - | |
| **Comorbidity** | | | |
| No | Ref | - | |
| Yes | 1.04(0.69–1.56) | - | |
| **Tobacco Use** | | | |
| No | Ref | Ref | |
| Yes | 1.60(1.22–2.12) | 1.54(0.95–2.48) | 0.117 |
| **Alcohol Use** | | | |
| No | Ref | Ref | |
| Yes | 1.5(1.07–2.11) | 0.93(0.51–1.69) | 0.909 |

likely to be cured from TB compared to the HIV negative patients, however this finding was not statistically significant (RR: 0.88; 95% C.I.: 0.69–1.11). The length of TB treatment was associated with TB cure. The results show that for each additional month on TB treatment there was a 3% significantly increased likelihood of TB cure (RR: 1.03; 95% C.I.: 1.01–1.06). Alcohol use and Tobacco smoking were not significantly associated with increased risk of treatment success.

Table 3 below present's results from the multinomial logistic regression model with two levels of outcomes 'dead and other'. We found that HIV status and age were associated with increased mortality among TB patients on treatment. The results show that HIV positive TB patients were nearly 4 times likely to die of TB compared to the HIV negative patients, and this finding was statistically significant (RRR: 3.73; 95% C.I.: 1.24–11.19). Furthermore, older age was associated with 5 times increased risk of dying among TB patients aged 50 years and above, however this was not statistically significant (RR: 4.99; 95% C.I. 0.88–28.50). In addition, duration of TB treatment was associated with reduced risk of dying. We found that for each additional month on TB treatment there was a 44% significant reduction in the risk of TB mortality (RR: 0.56; 95% C.I.: 0.47–0.66).

## 4. Discussion

This study determined the effect of HIV status and antiretroviral treatment on the treatment outcome of TB patients in a primary healthcare facility in rural South Africa. The study comprised of 282 males (55.7) and 224 females (44.3) with a median age of 32 (IQR: 23–43) years. The main outcomes of the study were categorized as cured or dead or other. This study found that 39.13% (198/506) of the patients were cured of Tuberculosis, of which 115 were HIV

**Table 3. Risk factors associated with death among TB patients presenting for care at Elim Clinic.**

| Factors | Univariate Analysis RRR (95% C.I.) | Adjusted Analysis RRR (95% C.I.) |
|---|---|---|
| **Duration on treatment** | 0.54 (0.46–1.13) | 0.56 (0.47–0.66) |
| **HIV Status** | | |
| Negative | Ref | Ref |
| Positive | 3.63 (1.46–9.02) | 3.73 (1.24–11.19) |
| **Age group** | | |
| <20 | Ref | Ref |
| 20–29 | 4.63 (0.98–21.97) | 2.40 (0.45–12.78) |
| 30–39 | 5.08 (1.07–24.13) | 2.18 (0.39–12.29) |
| 40–49 | 2.20 (0.35–13.71) | 1.46 (0.20–10.83) |
| 50+ | 8.51 (1.77–40.98) | 4.99 (0.88–28.50) |
| **Sex** | | |
| Female | Ref | Ref |
| Male | 0.65 (0.32–1.32) | 0.61 (0.26–1.46) |
| **Co-morbidity** | | |
| No | Ref | Ref |
| Yes | 1.83 (0.58–5.75) | 1.30 (0.29–5.80) |
| **Tobacco Use** | | |
| No | Ref | Ref |
| Yes | 0.51 (0.06–3.95) | 0.56 (0.06–4.96) |

*Other outcomes include. . .. Alive, transferred, defaulted etc

positive. Even though the patient's records did not clarify that TB patients completed a full course of treatment, our findings show that the treatment success rate may be estimated as below the WHO target of 90% [39]. The overall TB related mortality rate was 6.92% (n = 35 patients) and the study showed that the risk of mortality rate was significantly associated with HIV positive status. For instance, the mortality rate was higher among HIV positive patients (9.67%, i.e., 29 patients out of 300 patients) compared to HIV negative patients (2.91%, 6 patients out of 206 HIV negative patients). Longer treatment period was associated with a lowered risk of death; however, HIV positive status of a TB patient had no effect on the likelihood of TB cure when compared to HIV negative status of a TB patient.

In reviewing other studies conducted in rural Africa, HIV infection did not appear to have any effect on successful treatment of TB, and this could be due to lack of sufficient data in a study conducted in Ghana [40], however, the same study showed that HIV-positive status was associated with death [40]. Our study found a close association between HIV-positive status and increased risk of dying of TB like the Ghana study that showed a higher mortality rate among HIV/TB co-infected individuals [40]. Another study in Tanzania found an EPTB prevalence of 5.6% with a high mortality rate of 14.3% and a combined death/LTFU rate of 46.8% among people living with HIV [41]. Patients with EPTB not receiving ART and >45 years of age were at a higher risk for poor outcomes [41]. There were no differences in treatment success rates for HIV-positive TB patients on ART and HIV-negative in Kenya [42]. The similarities and differences between studies could be explained by the levels of ART uptake and TB diagnosis. Following a review several of these studies, our study finding is consistent with the study conducted in Kenya, owing to the ongoing increasing scale-up and uptake of ART among HIV positive individuals in both South Africa and Kenya. In Table 4 we show an ART uptake rate of 89.9%, compared to 61% in Kenya [42]. In contrast, 3.7% of ART uptake was

**Table 4. Characteristics of TB patients presenting to Elim Clinic on the basis of HIV status.**

| Characteristics | HIV Negative n(%) | HIV Positive n(%) | P-value |
|---|---|---|---|
| | 178 (37.2) | 300 (62.8) | |
| **Categories of TB** | | | |
| New | 161 (91.0) | 259 (86.3) | 0.395¶ |
| RAC | 8 (4.5) | 25 (8.3) | |
| RF | 5 (2.8) | 8 (2.7) | |
| RI | 3 (1.7) | 8 (2.7) | |
| **Pre-treatment sputum smear status** | | | |
| Negative | 27 (29.7) | 41 (30.4) | 0.910‡ |
| Positive | 64 (70.3) | 94 (69.6) | |
| **Type of Tuberculosis** | | | |
| Pulmonary | 124 (69.7) | 210 (70.0) | 0.938‡ |
| Extra-Pulmonary | 54 (30.3) | 90 (30.0) | |
| **Site of Tuberculosis** | | | |
| Bones/Joints | 1 (2.0) | 0 (0.0) | 0.047*¶ |
| Lymph Nodes | 0 (0.0) | 4 (4.5) | |
| Miliary | 4 (7.8) | 6 (6.7) | |
| Meningitis | 2 (3.9) | 17 (19.1) | |
| Primary | 12 (23.5) | 17 (19.1) | |
| Pleura | 12 (23.5) | 22 (24.7) | |
| Other organs | 20 (39.2) | 23 (25.8) | |
| **Receiving ART** | | | |
| No | 172 (100.0) | 30 (10.1) | <0.001*¶ |
| Yes | 0 (0.0) | 267 (89.9) | |
| **Co-trimoxazole Prophylaxis** | | | |
| No | 173 (100.0) | 22 (7.3) | <0.001*¶ |
| Yes | 0 (0.0) | 278 (92.7) | |
| **Outcome duration(days)** | 184 (176–203)** | 184 (173–209)* | 0.7547† |
| **Comorbidity** | | | |
| No | 155 (87.1) | 286 (95.3) | 0.001* |
| Yes | 23 (12.9) | 14 (4.7) | |
| **Regimen type at baseline** | | | |
| HRZE | 157 (88.7) | 280 (93.4) | 0.005*¶ |
| HR | 15 (8.5) | 10 (3.3) | |
| RHZ | 5 (2.8) | 3 (1.0) | |
| HRZES | 0 (0.0) | 7 (2.3) | |

N- Total number in each group; %—row percentages; Numbers may not add-up because of missing variables.

*Significant p-value;

**Median and Inter-quartile ranges (IQR); Test statistic based on ¶Fisher's Exact, ‡Chi-square and Wilcoxson †Rank-sum test.

HIV–Human Immunodeficiency Virus; ART–Antiretroviral therapy; Categories of TB treatment: RAC–retreatment after completion of a previous course without microscopic result; RF–Treatment failure: a patient who, while on treatment, remained or became again smear positive five months or later after commencing treatment; RI–Treatment after interruption: a patient whose treatment is interrupted for two or more months and who returned to the health service.

HRZE–Isoniazid, Rifampicin, Pyrazinamide and Ethambutol for New adult patients; HRZ–Isoniazid, Rifampicin and Pyrazinamide for New Paediatric patients; HRZES–Isoniazid, Rifampicin, Pyrazinamide, Ethambutol and Streptomycin for Retreatment patients.

recorded in Ghana [40] and 16% patients with extrapulmonary TB did not receive ART in Tanzania [41]. Aside from that, our study found 70.0% of patients to have pulmonary TB, compared to 36.9% in a Tanzanian study [41]. In fact, pulmonary TB is easier to diagnose than other types of TB. Previous data shows that Extrapulmonary TB is associated with poor TB outcome in HIV-infected people [43]. In our study low number of people were on ART at that time as the scaling up program was still in its infancy stage, hence difficult to make a meaningful analysis to assess its impact on TB outcome.

Age was reported as significantly associated with being treatment success. In our study, older patients were less likely to be treatment success and more likely to die of TB compared to younger patients. These results are in line with the study conducted in Zimbabwe where elderly patients had a higher risk of not being cured and high risk of dying [44]. Another study reported that older age patients are more likely to develop extra-pulmonary and atypical forms of TB disease that are often harder to diagnose than sputum smear-positive pulmonary tuberculosis [45]. Extra pulmonary and smear negative PTB have all previously shown to be associated with high TB mortality [43, 46, 47].

Existing evidence from previous meta-analysis of high-quality studies has shown that smoking tobacco was noted to influence the outcome of TB treatment as smokers had a likelihood of being cured compared with non-smokers (OR of 2.6 (95%CI 2.1–3.4) [48]. Further, the associations between smoking and TB mortality showed a pooled OR of 1.3 (95%CI 1.1–1.6) [48]. Another large systematic review showed that cigarette smoking was significantly linked with poor TB treatment outcomes [49]. This shows that evidence on the effect of smoking on TB outcomes remains inconclusive. The findings of our study however confirm what has been previously reported showing an association between smoking tobacco and TB cure. Smoking tobacco affects both innate and adaptive immunity, weakening the immunological defensive system in humans [50]. This appears to be the reason why smokers are at a higher risk of developing extra-pulmonary tuberculosis [51]. Apart from this, smoking tobacco increases the risk of mycobacterium TB infection as well as the development of tuberculosis in infected individuals [52]. However, it does not have a negative effect on cure among those on TB treatment.

A study done in in the KZN revealed that staff rotation challenges across different services have been contributed to skills insufficiency (80%); staff shortage (50%); high staff turnover, absenteeism, and staff personal preferences (30%) in the KZN [5]. Despite a substantial investment in the clinical expansion program in SA, sparsely populated rural areas are still constrained in geographic distribution and access to health services (seen at a lower PHC utilization rate) and this is a major challenge in provision HIV and TB services in rural areas [33]. In this context, integrated services could enhance and promote equity as they maximize the potential benefit for each facility visit for health care access [33]. Besides this, a large TB cohort in Western Cape, South Africa showed that concomitant diseases were not associated with TB mortality [53]. This has been clearly shown in our study. However, in the same study, factors such as 'treatment category' and 'comorbidity' were significantly associated with unsuccessful TB treatment [53].

Coronavirus disease 2019 (COVID-19) has a significant impact on TB services in addition to the previous TB challenges faced in rural areas in SA. A study conducted in a rural area in SA discovered a significant drop in total TB treatment enrolment from April to December 2020 when compared to April to December 2019, with the largest drop occurring in July 2020 (–26%) [54]. Data shows a decrease across all TB indicators in April, the month of the first COVID-19 lockdown, with positive TB tests experiencing the greatest decrease (–33%) [54]. TB testing and diagnosis decreased by 50% and 33%, respectively, during the COVID-19 lockdown in SA [55, 56]. Furthermore, the National Health Laboratory Service (NHLS) reported that the number of GeneXpert tests performed decreased by 26% between 2019 and 2020, and

the proportion of positive tests decreased by 18% in SA [57]. Given that the GeneXpert is a key diagnostic test for TB in HIV-positive people, the TB mortality rate in HIV-positive patients may be higher in rural areas during the COVID-19 pandemic. In addition to the findings of this study, COVID-19 may jeopardize the End TB strategy in rural area in SA.

The strength of our study is our sample size and its representativeness of the rural population, thereby minimizing selection bias. It was sufficient to estimate HIV/TB outcomes in rural primary care settings. Our study had several limitations that could lead to underestimation of HIV/TB outcomes. Retrospective cohort design uses records that have already been collected. This is an inherent weakness of the nature of use of existing medical records as data used is dependent on data documented in the records. We did not obtain the information on treatment completed outcome, HIV viral load and CD4 cells as this information was not available in the patients' medical records. As such, we were unable to accurately link all our patients' outcomes to these specific exposures. It is also possible that some of the patients included in our study were misclassified based on exposure or outcome status during analysis. Other study limitations included the fact that no patients were diagnosed nor treated for MDR-TB, which may limit the study's generalizability. "Additionally, 5.5% of the HIV status was missing, which might have hampered our understanding of the effect on HIV on mortality among people with TB. To overcome missing data challenges, we used complete case (or available case) analysis or listwise deletion approach, hence included in the final analysis only cases with complete data available.

## 5. Conclusion

In conclusion, HIV positive status, and length of treatment were significantly associated with TB treatment outcomes in this rural primary care facility. The risk of dying was higher among the HIV positive TB patients while duration on treatment was associated with increased likelihood of treatment success and reduced risk of mortality. Despite a higher ART coverage in general, HIV status was a strong predictor of the TB mortality in rural SA. However, the TB treatment success rate in rural SA was lower than the WHO target. This study could have a significant impact on the reinforcement of HIV/TB program integration in rural SA. To achieve the End TB strategy in rural SA, HIV/TB integration services needs to be reviewed to identify potential challenges in its implementation. Furthermore, various HIV/TB indicators should be reviewed to allow for strengthening of TB/HIV integration monitoring approach by the TB program.

## Supporting information

**S1 File. Data analysis commands.**
(XLS)

## Acknowledgments

The authors would like to thank Ugu Health District management for the permission to conduct the study and the staff of Elim Clinic for the support provided during the duration of this study.

## Author Contributions

**Conceptualization:** Peter S. Nyasulu, Jabulani Ncayiyana.

**Data curation:** Peter S. Nyasulu, Emery Ngasama, Lovemore N. Sigwadhi, Birhanu T. Ayele, Teye Umanah, Jabulani Ncayiyana.

**Formal analysis:** Emery Ngasama, Lovemore N. Sigwadhi, Lovelyn U. Ozougwu, Ruvimbo B. C. Nhandara, Birhanu T. Ayele, Teye Umanah.

**Investigation:** Peter S. Nyasulu, Emery Ngasama, Jacques L. Tamuzi, Birhanu T. Ayele, Jabulani Ncayiyana.

**Methodology:** Peter S. Nyasulu, Emery Ngasama, Jacques L. Tamuzi, Lovemore N. Sigwadhi, Birhanu T. Ayele, Jabulani Ncayiyana.

**Writing – original draft:** Peter S. Nyasulu, Jacques L. Tamuzi, Lovelyn U. Ozougwu, Ruvimbo B. C. Nhandara.

**Writing – review & editing:** Peter S. Nyasulu, Emery Ngasama, Jacques L. Tamuzi, Lovemore N. Sigwadhi, Lovelyn U. Ozougwu, Ruvimbo B. C. Nhandara, Birhanu T. Ayele, Teye Umanah, Jabulani Ncayiyana.

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
