## [Decision Letter · Decision Letter 0]

5 Aug 2021

PONE-D-21-05746

Effect of HIV status and antiretroviral treatment on treatment outcomes of tuberculosis patients in a rural primary healthcare clinic in South Africa

PLOS ONE

Dear Dr. Nyasulu,

Thank you for submitting your manuscript to PLOS ONE. After careful consideration, we feel that it has merit but does not fully meet PLOS ONE’s publication criteria as it currently stands. Therefore, we invite you to submit a revised version of the manuscript that addresses the points raised during the review process.

The reviewers have made a number of helpful suggestions that we hope will assist you in revision to meet PLOS ONE's publication criteria.

We look forward to receiving your revised manuscript.

Kind regards,

Janet E Rosenbaum, Ph.D.

Academic Editor

PLOS ONE

Journal Requirements:

2. In the ethics statement in the manuscript and in the online submission form, please provide additional information about the patient records used in your retrospective study, including: a) whether all data were fully anonymized before you accessed them; b) the date range (month and year) during which patients' medical records were accessed; c) the date range (month and year) during which patients whose medical records were selected for this study sought treatment. If the ethics committee waived the need for informed consent, or patients provided informed written consent to have data from their medical records used in research, please include this information.

 N/A

5. We note that Figure 1 in your submission contain map images which may be copyrighted. All PLOS content is published under the Creative Commons Attribution License (CC BY 4.0), which means that the manuscript, images, and Supporting Information files will be freely available online, and any third party is permitted to access, download, copy, distribute, and use these materials in any way, even commercially, with proper attribution. For these reasons, we cannot publish previously copyrighted maps or satellite images created using proprietary data, such as Google software (Google Maps, Street View, and Earth). For more information, see our copyright guidelines: http://journals.plos.org/plosone/s/licenses-and-copyright.

6. Please upload a copy of Supporting Information S1 Dataset which you refer to in your text on page 18.

Reviewers' comments:

Reviewer's Responses to Questions

**Comments to the Author**

1. Is the manuscript technically sound, and do the data support the conclusions?

Reviewer #1: Yes

Reviewer #2: Yes

2. Has the statistical analysis been performed appropriately and rigorously? 

Reviewer #1: No

Reviewer #2: Yes

3. Have the authors made all data underlying the findings in their manuscript fully available?

Reviewer #1: Yes

Reviewer #2: Yes

4. Is the manuscript presented in an intelligible fashion and written in standard English?

Reviewer #1: Yes

Reviewer #2: Yes

5. Review Comments to the Author

Reviewer #1: I wish to record my sincere appreciation to the authors for addressing this important topic. There is good insight to be gained from the research. However, my opinion is that the presentation style needs to be improved in order to allow a reader to gain proper insight into the presentation. I have raised a number of comments on the PDF that are required to be addressed

Reviewer #2: The abstract is well written. It contains the main findings and the conclusion of the study.

The background is comprehensive and well written. Although, it does not reflect on the decreasing TB cases that South Africa is experiencing.

The methodology is clear, well defined. The paper does not clearly define treatment cure, treatment completed. The researcher used the outcome allocated by the facility. I am not sure whether these outcomes were checked. Sometimes patients who fulfill the cure criteria are captured as completed. It is also important to highlight that the outcomes are allocated mainly on the basis of negative TB smear microscopy. TB microscopy has a low sensitivity. TB culture is not done routinely in susceptible TB patients.

My other issue is the choice of comparing cure or death. TB programme targets are based on treatment success rate (Cure and Completion). I agree that it is better to have a higher cure rate although for several reasons I have noted over the years the final sputum is often not collected hence the outcome will be "treatment completed". Also Table 1 indicates that 30 % of the cohort had a negative smear microscopy. Such patients may not have a "cure" outcome. Favorable outcomes include cure and completion rate. I am not sure what is the motivation for having cure, death and other outcomes. It would have been useful to briefly explain why are there more treatment completion than treatment cure in this facility. Data analysis is very clear except the issue of separating cure and completion. On page 11 of the manuscript, the Authors stated that among tobacco users the cure rate was for TB was 60 % and it was only 37 % among the non-tobacco users. In the conclusion, it is stated that the use of tobacco appeared to decrease the TB cure. It is also said that the cure rate was below district and provincial targets. To the best of my knowledge, there are no separate targets between cure and completion. Targets are based on Treatment Success Rate.

6. PLOS authors have the option to publish the peer review history of their article (what does this mean?). If published, this will include your full peer review and any attached files.

Reviewer #1: **Yes: **Ozayr Mahomed

Reviewer #2: **Yes: **Dr Norbert NDJEKA, Director DR-TB, TB & HIV, National Department of Health, Republic of South Africa.

---

## [Author Response · Author response to Decision Letter 0]

24 Jan 2022

Reviewers’ responses to reviewers

Title: Effect of HIV status and antiretroviral treatment on treatment outcomes of tuberculosis patients in a rural primary healthcare clinic in South Africa

Journal Requirements:

Comment 1. Please ensure that your manuscript meets PLOS ONE's style requirements, including those for file naming. The PLOS ONE style templates can be found at 

Response: Thanks, we have revised the manuscript in accordance with the PLOS One formatting sample.

Comment 2. In the ethics statement in the manuscript and in the online submission form, please provide additional information about the patient records used in your retrospective study, including a) whether all data were fully anonymized before you accessed them; b) the date range (month and year) during which patients' medical records were accessed; c) the date range (month and year) during which patients whose medical records were selected for this study sought treatment. If the ethics committee waived the need for informed consent, or patients provided informed written consent to have data from their medical records used in research, please include this information.

Response: Thanks for the comments and suggestions. Data collection for this study commenced from the 1st of September 2016 and ended on 31st of October 2016. The data clerk was contracted for a 2-month period to enter as well as do quality control of the data entered. Data were abstracted directly from registers of TB patients who were initiated on TB treatment from 1 January 2013 to 31 December 2015. Furthermore, TB patients registered in the TB register in 2012 and died in 2013 were also included in the data extraction. (Line 147 to 152 Page 8). The data entry clerk was under strict instruction to enter data anonymously from the TB registers onto the database. No identifiers were extracted from the TB registers. It should be noted that this data were extracted in 2016 of patients who received service for TB treatment almost 3 years previously. As a result it was not possible to get patient consent as these were historical records and patients had been out of the system for 3 years already by the time the study was being conducted, so we had to seek waiver of consent to use the data as we believed that we could generate informative data that would shape policy and treatment guidelines in the management of TB at a time when expanding access to TB/HIV collaborative treatment was actively being rolled out by the department of health as a successful model of clinical care to minimise poor outcomes. Ethical approval was granted by the Monash University Human Research Ethics Committee (approval number: CF16/2803- 2014001548). Permission to conduct the study at Elim Primary Health Care Clinic was obtained from the Director of in Ugu Health District, KwaZulu Natal. 

Comment 3. Thank you for stating the following financial disclosure: 

Response: N/A

a) Please clarify the sources of funding (financial or material support) for your study. List the grants or organizations that supported your study, including funding received from your institution. N/A

b) State what role the funders took in the study. If the funders had no role in your study, please state: “The funders had no role in study design, data collection and analysis, decision to publish, or preparation of the manuscript.” N/A

c) If any authors received a salary from any of your funders, please state which authors and which funders. N/A

Response: In track change document on page 26, Line 492. Thanks, we state this: “Financial Disclosure: The author(s) received no specific funding for this work”.

Comment 4. In your Data Availability statement, you have not specified where the minimal data set underlying the results described in your manuscript can be found. PLOS defines a study's minimal data set as the underlying data used to reach the conclusions drawn in the manuscript and any additional data required to replicate the reported study findings in their entirety. All PLOS journals require that the minimal data set be made fully available. For more information about our data policy, please see http://journals.plos.org/plosone/s/data-availability.

Response: In track change document on page 23, lines 423-424. Thanks, Dataset is available upon request by submitting the request form to the corresponding author (email: pnyasulu@sun.ac.za).

Comment 5. We note that Figure 1 in your submission contain map images which may be copyrighted. All PLOS content is published under the Creative Commons Attribution License (CC BY 4.0), which means that the manuscript, images, and Supporting Information files will be freely available online, and any third party is permitted to access, download, copy, distribute, and use these materials in any way, even commercially, with proper attribution. For these reasons, we cannot publish previously copyrighted maps or satellite images created using proprietary data, such as Google software (Google Maps, Street View, and Earth). For more information, see our copyright guidelines: http://journals.plos.org/plosone/s/licenses-and-copyright.

USGS EROS (Earth Resources Observatory and Science (EROS) Centre) (public domain): http://eros.usgs.gov/#

Response: Thanks for this comment. The map was created using a shapefile from the Africa map library software, which was then imported into the Quantum Geographic information system (QGIS) software for georeferencing of the Ugu district. Figure 1 is licensed under a CC BY 4.0

Comment 6. Please upload a copy of Supporting Information S1 Dataset which you refer to in your text on page 18.

Response: In track change document on page 21 and lines 399. Thanks, we have uploaded the raw data for the study.

Reviewer #1

Comment 1: although this statement maybe correct, it cannot be a conclusion at the level of the abstract as the reader has no comparator.

Response: In track change document on page 3 and lines 46 to 49. The conclusion is now: “when compared HIV positive and HIV negative status, antiretroviral treatment had no effect on the likelihood of TB cure in rural primary care. The TB mortality rate in HIV positive patients, on the other hand, was higher than in HIV negative patients”.

Comment 2: based on this what is your recommendation.

Response: In track change document on page 3 and lines 49 to 50. We recommend this: “Various HIV/TB indicators should be reviewed, and gaps filled in order to achieve the “End TB strategy” in rural South Africa”.

Comment 3: comma

Response: In track change document on page 4 and line 64, the comma has been inserted.

Comment 4: I suggest that paragraph 2 is integrated at this point to improve the flow and readability of the manuscript.

Response: In track change document on page 4 and lines 65-71. Thanks, paragraph 2 has been integrated as suggested.

Comment 5: I think that this sentence is redundant information.

Response: In track change document on page 4 and lines 75-77. Thanks, this sentence has been written more clearly in the background.

Comment 6: There are two concepts being discussed in this sentence- decrease in mortality and increased incidence. however, the sentence needs multiple readings to make sense. therefore, I suggest that the sentence is separated.

Response: In track change document on page 4 and lines 63-66. We have separated the two concepts. It now reads: “The increase in incidence is also attributed to the development of multidrug resistant (MDR) and extremely drug resistant (XDR) strains of Mycobacterium tuberculosis [34]. Both MDR/XDR-TB are the causes of high TB mortality [4]”.

Comment 7: This sentence appears out of sync as the previous discussion only relate to mortality and this sentence intends on providing reason for increased risk for TB.

Response: The sentence has been deleted and removed.

Comment 8: Which countries as the authors quoting facility based postmortem findings.

Response: In track change document on pages 4-5 and lines 75-77. Thanks, we have listed the countries reporting post-mortem TB rate. It now reads: “such as South Africa [9-11], Botswana [11, 12], Zimbabwe [11, 13, 14], Mozambique [11, 15], Uganda [11, 16] and Kenya [11, 17]”. 

Comment 9: I am sure that the WHO global report on TB provides data in resource limited countries.

Response: In track change document on page 5 and lines 77. Thanks, we have provided TB data in Africa. “In contrast, the WHO reported 16% of HIV/TB related death in Africa [1]”.

Comment 10: please update this data as later information is available.

Response: In track change document on pages 5 and lines 80-81. Thanks, this data has been updated. “TB accounted for the third highest number of deaths in 2018 (6 %; n = 454 014)”

Comment 11: specific data for Ugu district

Response: In track change document on page 5 and lines 87-88. “Ugu district reported 60.5% of TB/HIV co-infection rate [25]”.

Comment 12: provide information on South Africa antiretroviral treatment programme in terms of number of patients on treatment and accessibility to treatment.

Response: In track change document on pages 6 and lines 89-90. Thanks, we have reported this: “According to recent data, 90% of people are aware of their HIV status of these 68% are on antiretroviral therapy (ART) of which 87% are virally suppressed in SA [26]”. 

Comment 13: this paragraph is out of sync in building the case. it may be more suited to a discussion

Response: In track change document on pages 19-20, lines 360-367. Thanks, this paragraph has been moved to the discussion section.

Comment 14: with a view to?

Response: In track change document on page 7 and lines 141. Thanks, we state as follows: “with a view to Ugu district”.

Comment 15: the author needs to separate study design, population and study setting

Response: In track change document on pages 8-9 and lines 122-142.Thanks, we have separated the study design, population and study setting. It is now reads:

“Study design 

This retrospective cohort study of TB patients initiated on TB treatment was conducted from 1 January 2013 to 31 December 2015. 

Study population

We included all patients diagnosed with TB irrespective of their age and HIV status in the study. Further, we registered patients that were in the TB register in 2012 and died in 2013. We also included as well as those that died or survived during the treatment period. Patients with unknown outcome were from the study or those with incomplete record were excluded. Basic demographic information including the age, gender, co-morbidities, tobacco Use, alcohol Use, substance use and duration on treatment were collected.

Study setting

The Ugu district is located in the rural KwaZulu-Natal province (Figure 1). Ugu district has a population of 733 228 people [37]. During the study period, Ugu district had the highest HIV prevalence and TB incidence of any district in KZN, 41.7% and 1096 per 100,000 people, respectively [37]. In terms of infectious TB (pulmonary smear-positive), Ugu ranks 12th, with 325 cases per 100,000 people, which is higher than the country's average of 208 cases per 100,000 people [37]. Elim clinic, a primary health care facility was selected based on convenience, the study goal and the availability of information on HIV and TB infections”.

Comment 16: this is about uGu- what about the specific area of your study- what is catchment population, what is outpatient headcount? what proportion or incidence of TB per year over the study period?

Response: In track change document on pages 7 and lines 136-142. Thanks, we have stated the following: “Ugu district has a population of 733 228 people [37]. During the study period, Ugu district had the highest HIV prevalence and TB incidence of any district in KZN, 41.7% and 1096 per 100,000 people, respectively [37]. In terms of infectious TB (pulmonary smear-positive), Ugu ranks 12th, with 325 cases per 100,000 people, which is higher than the country's average of 208 cases per 100,000 people [37]. Elim clinic, a primary health care facility was selected based on convenience, the study goal and the availability of information on HIV and TB infections”.

Comment 17: was provincial authorization obtained? I am confused by the term Director of Health as no such position exists in the organogram.

Response: Thanks for this comment. Authorisation to access clinic records was granted by the Ugu Health District Manager. Apologies for the incorrect use of word ‘Director’.

Comment 18: it would be good to also provide the socio-demographic profile of patients first before the HIV status of patients.

Response: In track change document on page 12and line 240. Thanks, before the HIV status table (Table 2), we have provided a socio-demographic table (Table 1).

Comment 19: p values determination- and if any of the information is statistically significant

Response: In track change document on page 15, table 3, line 277. Thanks, p-values have been provided. Duration on treatment and age-group were significant predictors of cure.

Comment 20: were the values statistically significant?

Response: In track change document on page 15, line 277. “The 95% CI shows that the values obtained were statistically significant”.

Comment 21: to evaluate TB outcomes in HIV positive patients in rural primary healthcare in South Africa

Response: In track change document on page 17, lines 300. Thanks, it now reads: “This study evaluated the effect of HIV status and antiretroviral treatment on the treatment outcome of TB patients in a primary healthcare facility in rural South Africa”.

Comment 22: what about findings from your analytical study?

Response: In track change document on page 17, lines 312-317. We have reported the findings from the analytical study as follows: “Longer treatment periods were associated with a lower risk of death in both the bivariate and covariate log-binomial regression models. Furthermore, in bivariate and covariate analysis, younger ages had a lower likelihood of being cured than older ages. However, HIV positive status of a TB patient had no effect on the likelihood of TB cure when compared to HIV negative status of a TB patient”.

Comment 23: please clarify this- is this initiation of treatment or treatment duration

Response: In track change document on page 17, lines 315-317. It now reads “However, HIV positive status of a TB patient had no effect on the likelihood of TB cure when compared to HIV negative status of a TB patient”.

Comment 24: how does this study relate to your findings. what is the plausible explanation for the similarities and differences?

Response: In track change document on page 18, lines 329-340. Thanks, the plausible explanation for the similarities and differences is stated as follows: “The similarities and differences between studies could be explained by ART uptake percentage and TB diagnosis. Following a review of these studies, the findings of our study was consistent with the study conducted in Kenya, owing to the ongoing scale-up and uptake of ART programs in both South Africa and Kenya. Table 2 of our study revealed an ART uptake rate of 89.9%, compared to 61% in Kenya [42]. In contrast, 3.7% of ART uptake was recorded in Ghana [40] and 16% patients with extrapulmonary TB did not receive ART in Tanzania [41]. Aside from that, our study found 70.0% of patients to have pulmonary TB, compared to 36.9% in a Tanzanian study [41]. In fact, pulmonary TB is easier to diagnose than other types of TB. Previous data shows that “Extrapulmonary TB is associated with poor TB outcome in HIV-infected people [43]”.

Comment 25: is better adherence to treatment the only plausible explanation? if you review the data. It is noticed that age > 50 years also had a very high likelihood of mortality and therefore, i question your statement provided.

Response: In track change document on page 19, lines 349-353. Thanks, we have provided the following statement:” Another study has shown that older ages are more likely to develop extra-pulmonary and atypical forms of TB disease that are often harder to diagnose than conventional sputum smear-positive pulmonary tuberculosis [45]. Extra pulmonary and smear negative PTB were associated with high TB mortality [43, 46, 47].”

Comment 26: what is the plausible explanation for tobacco and TB treatment outcomes?

Response: In track change document on page 19, lines 362-366. The discussion part is improved as follows: “Smoking tobacco affects both innate and adaptive immunity, weakening the immunological defensive system in humans [50]. This appears to be the reason why smokers are at a higher risk of developing extra-pulmonary tuberculosis [51]. Apart from this, smoking tobacco increases the risk of mycobacterium TB infection as well as the development of tuberculosis in infected individuals [52]”.

Comment 27: please discuss the study limitations

Response: In track change document on page 20, lines 381-388. We have included the study strengths and limitations. It now read: “The strength of our study is our sample size which was representative of the study population thereby minimizing selection bias. This is substantial to estimate HIV/TB outcomes in rural settings. Our study has several limitations that could lead to underestimation HIV/TB outcomes. Retrospective cohort design uses records that have already been collected and we did not obtain the information on treatment completed outcome, HIV viral load and CD4 cells. As the fact, we were unable to accurately link all our patients to the various outcomes. It is also possible that some of the patients included in our study misclassified in the current analysis”.

Comment 28: this is a repetition of results without adequate considering for the overall aim and why the study was done?

Response: In track change document on page 21, lines 392-395, 400-404. Thanks, we have improved the conclusion as follows: “In conclusion, HIV positive status and antiretroviral treatment had no effect on the likelihood of TB cure in rural primary care when compared to HIV negative patients. However, the TB mortality rate in HIV positive patients was higher than in HIV negative patients”. And further recommendation: “TB mortality in rural SA. Furthermore, the TB success rate in rural SA may be lower than the WHO target. This study could have a significant impact on the HIV/TB program in rural SA. To achieve the End TB strategy in rural SA, various HIV/TB indicators should be reviewed, and gaps filled”.

Reviewer #2: 

Comment 1: The abstract is well written. It contains the main findings and the conclusion of the study. The background is comprehensive and well written. Although, it does not reflect on the decreasing TB cases that South Africa is experiencing.

Response: In track change document on page 5-6, lines 103-108. Thanks, we have improved the background as follows:” TB incidence and mortality are declining in SA [7]. Data from a well-characterized rural SA population with high HIV prevalence and TB incidence demonstrated considerable spatial heterogeneity in people with recently-diagnosed TB and has shown that every percentage increase in ART coverage was associated with a 2% decrease in the odds of recently-diagnosed TB [23]”.

Comment 2: The methodology is clear, well defined. The paper does not clearly define treatment cure, treatment completed.

Response: In track change document on pages 8, lines 160-167. We have referred to the WHO definitions. Cured: A pulmonary TB patient with bacteriologically confirmed TB at the beginning of treatment who was smear or culture negative in the last month of treatment and on at least one previous occasion. Treatment completed: A TB patient who completed treatment without evidence of failure but with no record to show that sputum smear or culture results in the last month of treatment and on at least one previous occasion were negative either because tests were not done or because results are unavailable [37].

Comment 3: The researcher used the outcome allocated by the facility. I am not sure whether these outcomes were checked. Sometimes patients who fulfil the cure criteria are captured as completed. It is also important to highlight that the outcomes are allocated mainly on the basis of negative TB smear microscopy. TB microscopy has a low sensitivity. TB culture is not done routinely in susceptible TB patients.

Response: In track change document on page 20, lines 384-388. Thanks, we have addressed this issue in the study weaknesses. As a retrospective cohort study, we conducted the records review of the patients. As a matter of fact, we were unable to accurately link all of our patients to the various outcomes due to missing data. It is also possible that some of the patients included in our study might have been misclassified in the current analysis.

Comment 5: My other issue is the choice of comparing cure or death. TB programme targets are based on treatment success rate (Cure and Completion). I agree that it is better to have a higher cure rate although for several reasons I have noted over the years the final sputum is often not collected hence the outcome will be "treatment completed".

Response: Thanks for the observation, as previously stated, retrospective cohort design used records that have already been collected and we did not obtain the information on treatment completed from the records.

Comment 6: Also, Table 1 indicates that 30 % of the cohort had a negative smear microscopy. Such patients may not have a "cure" outcome. Favorable outcomes include cure and completion rate. I am not sure what is the motivation for having cure, death, and other outcomes. It would have been useful to briefly explain why are there more treatment completion than treatment cure in this facility. 

Response: In track change document on page 9, lines 160-163. Thanks, “cured” is defined as a pulmonary TB patient with bacteriologically confirmed TB at the beginning of treatment in an individual who was smear or culture negative in the last month of treatment and on at least one previous occasion. So, people may complete TB treatment without necessarily being cured. In this study we used these outcomes because they are key TB Program indicators as defined by the National TB Control Program. The focus of our study was to assess the effect of HIV status and antiretroviral treatment on TB treatment outcomes including cure. Our data has shown that HIV status and antiretroviral treatment did not have effect on TB cure but on TB mortality. We suggest that further studies are warranted to help explain low TB cure in this setting.

Comment 7: Data analysis is very clear except the issue of separating cure and completion. On page 11 of the manuscript, the Authors stated that among tobacco users the cure rate was for TB was 60 % and it was only 37 % among the non-tobacco users. In the conclusion, it is stated that the use of tobacco appeared to decrease the TB cure. It is also said that the cure rate was below district and provincial targets. To the best of my knowledge, there are no separate targets between cure and completion. Targets are based on Treatment Success Rate.

Response: Thanks for the observations, In track change document on page 19-20, Table 4. We separated cure and completion because they are two separate National TB Control Program treatment outcomes. The overall cure rate was 39.13% which falls below district and provincial targets based on South African District Health Barometer. Treatment Success Rate is the sum of cured and treatment completion rates.

---

## [Decision Letter · Decision Letter 1]

30 May 2022

PONE-D-21-05746R1Effect of HIV status and antiretroviral treatment on treatment outcomes of tuberculosis patients in a rural primary healthcare clinic in South AfricaPLOS ONE

Dear Dr. Nyasulu,

Thank you for submitting your manuscript to PLOS ONE. After careful consideration, we feel that it has merit but does not fully meet PLOS ONE’s publication criteria as it currently stands. Therefore, we invite you to submit a revised version of the manuscript that addresses the points raised during the review process.

We look forward to receiving your revised manuscript.

Kind regards,

Janet E Rosenbaum, Ph.D.

Academic Editor

PLOS ONE

Journal Requirements:

Reviewers' comments:

Reviewer's Responses to Questions

**Comments to the Author**

1. If the authors have adequately addressed your comments raised in a previous round of review and you feel that this manuscript is now acceptable for publication, you may indicate that here to bypass the “Comments to the Author” section, enter your conflict of interest statement in the “Confidential to Editor” section, and submit your "Accept" recommendation.

Reviewer #1: All comments have been addressed

Reviewer #3: (No Response)

2. Is the manuscript technically sound, and do the data support the conclusions?

Reviewer #1: Partly

Reviewer #3: Partly

3. Has the statistical analysis been performed appropriately and rigorously? 

Reviewer #1: Yes

Reviewer #3: Yes

4. Have the authors made all data underlying the findings in their manuscript fully available?

Reviewer #1: No

Reviewer #3: Yes

5. Is the manuscript presented in an intelligible fashion and written in standard English?

Reviewer #1: Yes

Reviewer #3: Yes

6. Review Comments to the Author

Reviewer #1: The aurthors have addressed most of the substantive points of the previous review. I have highlighted certain areas on the re-submission that need to be looked at again. However the aurthors have data data form more than 5 years ago and do nt provide a critical discussion of how this old data still stands currently with the growth and changes in the program. Is it not likely that a greater saturation in terms of ART ha s improved TB outcomes or there is a worsening due to COVID 19.

Reviewer #3: Line 126: would it have been helpful to distinguish pediatric from adult participants given likely baseline differences in HIV and TB risk?

Line 182: what proportion of data were missing and how were missing data accounted for?

Was HIV status confirmed on the basis of seropositivity using chart abstraction? If based on self-report, may result in measurement bias, possibly underestimating true HIV prevalence.

Use of TB cure definition using smear negativity may miss cases that are smear negative and go on to become culture-positive. Use of completed TB definition may include persons who were actually cured following treatment.

Based on baseline TB regimens, it appears none of the patients were being treated for multidrug resistant TB (which is prevalent in South Africa); this may limit generalizability/external validity.

Was matching on covariates between HIV positive and HIV negative participants considered?

7. PLOS authors have the option to publish the peer review history of their article (what does this mean?). If published, this will include your full peer review and any attached files.

Reviewer #1: **Yes: **Ozayr Mahomed

Reviewer #3: No

---

## [Author Response · Author response to Decision Letter 1]

20 Jul 2022

Reviewers’ responses to reviewers

Title: Effect of HIV status and antiretroviral treatment on treatment outcomes of tuberculosis patients in a rural primary healthcare clinic in South Africa

Journal Requirements:

Comment 1: Please review your reference list to ensure that it is complete and correct. If you have cited papers that have been retracted, please include the rationale for doing so in the manuscript text or remove these references and replace them with relevant current references. Any changes to the reference list should be mentioned in the rebuttal letter that accompanies your revised manuscript. If you need to cite a retracted article, indicate the article’s retracted status in the References list and include a citation and full reference for the retraction notice.

Response 1: Thank you for this guidance. In the tracking manuscript, the reference 10, 12, 14, 16, 17, 30, and 31 have been revised according to PLOS One submission guideline.

Reviewer #1: 

Comment 1: The authors have addressed most of the substantive points of the previous review. I have highlighted certain areas on the re-submission that need to be looked at again. However, the authors have data from more than 5 years ago and do not provide a critical discussion of how this old data still stands currently with the growth and changes in the program. Is it not likely that a greater saturation in terms of ART has improved TB outcomes or there is a worsening due to COVID-19.

Response 1: Thanks for this valuable comment. In the tracking document lines 382-396, the following has been highlighted: “Coronavirus disease 2019 (COVID-19) has a significant impact on TB services in addition to the previous TB challenges faced in rural areas in SA. A study conducted in a rural area in SA discovered a significant drop in total TB treatment enrolment from April to December 2020 when compared to April to December 2019, with the largest drop occurring in July 2020 (–26%) [54]. Data shows a decrease across all TB indicators in April 2020, the month of the first COVID-19 lockdown, with positive TB tests experiencing the greatest decrease (–33%) [54]. TB testing and diagnosis decreased by 50% and 33%, respectively, during the COVID-19 lockdown in SA [55, 56]. Furthermore, the National Health Laboratory Service (NHLS) reported that the number of GeneXpert tests performed decreased by 26% between 2019 and 2020, and the proportion of positive tests decreased by 18% in SA [57]. Given that the GeneXpert is a key diagnostic test for TB in HIV-positive people, the TB mortality rate in HIV-positive patients may be higher in rural areas during the COVID-19 pandemic. In addition to the findings of this study, COVID-19 may jeopardize the End TB strategy in rural area in SA.”.

Reviewer #2: 

Comment 1: on what premise is this conclusion derived to be presented in the abstract

Response 1: Thanks for the observation. In tracking manuscript line 46-49, we have changed the conclusion as follows: “When compared between HIV status, HIV positive TB patients were more likely to have unsuccessful treatment outcome in rural primary care”. Antiretroviral treatment seems to have had no effect on the likelihood of TB treatment success in rural primary care. The TB mortality rate in HIV positive patients, on the other hand, was higher than in HIV negative patients emphasizing the need for enhanced integrated management of HIV/TB in rural South Africa through active screening of TB among HIV positive individuals and early access to ART among HIV positive TB cases.

Comment 2: these results have not been presented in abstract

Response 2: Thanks for highlighting this. In the tracking manuscript line 42-43, “Furthermore, HIV positive patients had a higher mortality rate (9.67%) than HIV negative patients (2.91%)”.

Comment 3: at this point this has no bearing with the results in abstract or link to introduction

Response 3: Thanks for this comment, the conclusion has been revised based on above comment 1 and 2.

Comment 4: the term evaluate has a deeper meaning than just an epidemiological study as conducted by the authors

Response 4: Thanks. In the tracking manuscript line 118, It reads: “Therefore, this retrospective study has been undertaken to determine….”

Comment 5: geo spatial map of Ugu and the key TB indicators in comparison to other districts in the Province will be beneficial for external readers to picture the burden in Ugu

Response 5: Thanks for the guidance. Figure 1 has been designed according to this recommendation. 

Comment 6: out of sync

Response 6: Thanks for this observation. In the tracking manuscript, line 145, the title has been aligned.

Comment 7: based on what you have provided you have not evaluated you have only determined factors associated with

Response 7: Thanks for the observation. In the tracking manuscript, line 308, we have replaced “evaluated” to “determined”.

Comment 8: please clarify what HIV status is

Response 8: Thanks for this advice. Line 326, we have specified as follows: “HIV positive status”.

Comment 9: what is recent

Response 9: Thanks for this point seeking clarity. Line 371, “recent” has been removed from this sentence with appropriate edit.

Reviewer #3: 

Comment 1: Line 126: would it have been helpful to distinguish pediatric from adult participants given likely baseline differences in HIV and TB risk?

Response 1: Thanks for this observation. In the tracking document line 223-224: “Among them, only 13% (66/506) were children”. We do not believe that removing it will change the outcome.

Comment 2: Line 182: what proportion of data were missing and how were missing data accounted for?

Response 2: Thanks for this question. In the tracking document line 409-415: “Additionally, 5.5% of the HIV status was missing, which might have hampered our understanding of the effect on HIV on mortality among people with TB. To overcome missing data challenges, we used complete case (or available case) analysis or listwise deletion approach, hence include in the final analysis only cases with complete data available.

Comment 3: Was HIV status confirmed based on seropositivity using chart abstraction? If based on self-report, may result in measurement bias, possibly underestimating true HIV prevalence.

Response 3: Thanks for this comment. As part of a retrospective study design, HIV status was confirmed using medical records available to ascertain objectiveness.

Comment 4: Use of TB cure definition using smear negativity may miss cases that are smear negative and go on to become culture positive. Use of completed TB definition may include persons who were cured following treatment.

Response 4: Thanks for this comment. Given that the study outcome includes “cured” and “treatment completion”, we have added the following definition: “Treatment success includes the sum of cured and treatment completed [38]” (Line 168-169). “Cured” has been replaced by “treatment success” in the following line 49, 168, 172, 173, 226, 228, 231, 233, 232, 233, 235-238, 280, 286, 347, 348, and 422.

Comment 5: Based on baseline TB regimens, it appears none of the patients were being treated for multidrug resistant TB (which is prevalent in South Africa); this may limit generalizability/external validity.

Response 5: Thank for this valuable comment. Line 409-411. We have added the following: “Other study limitations included the fact that no patients were diagnosed nor treated for MDR-TB at this clinic, which may limit the study's generalizability”.

Comment 6: Was matching on covariates between HIV positive and HIV negative participants considered?

Response 6: Thanks for this question. Since HIV status was an important covariate in the statistical modelling, we did not perform matching of variables.

---

## [Decision Letter · Decision Letter 2]

31 Aug 2022

Effect of HIV status and antiretroviral treatment on treatment outcomes of tuberculosis patients in a rural primary healthcare clinic in South Africa

PONE-D-21-05746R2

Dear Dr. Nyasulu,

We’re pleased to inform you that your manuscript has been judged scientifically suitable for publication and will be formally accepted for publication once it meets all outstanding technical requirements.

Kind regards,

Janet E Rosenbaum, Ph.D.

Academic Editor

PLOS ONE

Additional Editor Comments (optional):

Reviewers' comments:

Reviewer's Responses to Questions

**Comments to the Author**

1. If the authors have adequately addressed your comments raised in a previous round of review and you feel that this manuscript is now acceptable for publication, you may indicate that here to bypass the “Comments to the Author” section, enter your conflict of interest statement in the “Confidential to Editor” section, and submit your "Accept" recommendation.

Reviewer #1: All comments have been addressed

Reviewer #3: All comments have been addressed

2. Is the manuscript technically sound, and do the data support the conclusions?

Reviewer #1: Yes

Reviewer #3: Yes

3. Has the statistical analysis been performed appropriately and rigorously? 

Reviewer #1: Yes

Reviewer #3: Yes

4. Have the authors made all data underlying the findings in their manuscript fully available?

Reviewer #1: No

Reviewer #3: Yes

5. Is the manuscript presented in an intelligible fashion and written in standard English?

Reviewer #1: Yes

Reviewer #3: Yes

6. Review Comments to the Author

Reviewer #1: All points addressed and quality of manuscript is enhanced. I am of teh opinion it is suitable for publication

Reviewer #3: (No Response)

7. PLOS authors have the option to publish the peer review history of their article (what does this mean?). If published, this will include your full peer review and any attached files.

Reviewer #1: **Yes: **Ozayr Mahomed

Reviewer #3: No

---

## [Editor Report · Acceptance letter]

12 Sep 2022

PONE-D-21-05746R2 

Effect of HIV status and antiretroviral treatment on treatment outcomes of tuberculosis patients in a rural primary healthcare clinic in South Africa 

Dear Dr. Nyasulu:

I'm pleased to inform you that your manuscript has been deemed suitable for publication in PLOS ONE. Congratulations! Your manuscript is now with our production department. 

Kind regards, 

on behalf of

Dr. Janet E Rosenbaum 

Academic Editor

PLOS ONE